# Effectiveness of Acute Malnutrition Treatment at Health Center and Community Levels with a Simplified, Combined Protocol in Mali: An Observational Cohort Study

**DOI:** 10.3390/nu14224923

**Published:** 2022-11-21

**Authors:** Suvi T. Kangas, Bethany Marron, Zachary Tausanovitch, Elizabeth Radin, Josiane Andrianarisoa, Salimou Dembele, Césaire T. Ouédraogo, Issa Niamanto Coulibaly, Marie Biotteau, Bareye Ouologuem, Soumaila Daou, Fatoumata Traoré, Issiaka Traoré, Marc Nene, Jeanette Bailey

**Affiliations:** 1International Rescue Committee, New York, NY, USA; 2International Rescue Committee, Bamako, Mali; 3Nutrition Division, Ministry of Health, Bamako, Mali; 4Nara District Health Unit, Nara, Mali; 5United Nations Children’s Fund, Bamako, Mali

**Keywords:** acute malnutrition, combined protocol, community-based management of acute malnutrition, mid-upper-arm circumference, simplified protocol, wasting

## Abstract

A simplified, combined protocol was created that admits children with a mid-upper-arm circumference (MUAC) of <125 mm or edema to malnutrition treatment with ready-to-use therapeutic food (RUTF) that involves prescribing two daily RUTF sachets to children with MUAC < 115 mm or edema and one daily sachet to those with 115 mm ≤ MUAC < 125 mm. This treatment was previously shown to result in non-inferior programmatic outcomes compared with standard treatment. We aimed at observing its effectiveness in a routine setting at scale, including via delivery by community health workers (CHWs). A total of 27,800 children were admitted to the simplified, combined treatment. Treatment resulted in a 92% overall recovery, with a mean length of stay of 40 days and a mean RUTF consumption of 62 sachets per child treated. Among children admitted with MUAC < 115 mm or edema, 87% recovered with a mean length of stay of 55 days and consuming an average of 96 RUTF sachets. The recovery in all sub-groups studied exceeded 85%. Treatment by CHWs resulted in a similar (94%) recovery to treatment by formal healthcare workers (92%). The simplified, combined protocol resulted in high recovery and low RUTF consumption per child treated and can safely be adopted by CHWs to provide treatment at the community level.

## 1. Introduction

Acute malnutrition is a condition currently diagnosed among children 6–59 months of age with a low mid-upper-arm circumference (MUAC), a low weight-for-height z-score (WHZ) and/or the presence of bilateral pitting edema [1]. While the combined global estimate for the prevalence when applying the three criteria is lacking, in 2020, 45 million children were estimated to be acutely malnourished using WHZ alone at any time [2].

The treatment of acute malnutrition is currently separated into different programs according to the severity of the condition; severe acute malnutrition (SAM) is treated in outpatient therapeutic feeding programs and moderate acute malnutrition (MAM) is treated in supplementary feeding programs [3]. These programs run parallel to each other and often treat patients on different days, sometimes at different sites. Each program uses its own nutritional treatment product with its own supply chain. Yet, the condition itself can be seen as a continuum from moderate to severe [4,5,6]. 

The two nutritional products used to treat SAM and MAM have very similar nutritional content with primarily the source of the protein differing [7,8]. However, the dosage and the purpose of the two nutritional products are very different. Ready-to-use therapeutic food (RUTF) is designed to cover all the nutritional needs of a child recovering from SAM and is prescribed according to the weight of the child. Ready-to-use supplementary food is used as a nutritional supplement to normal food with one sachet prescribed daily per child with MAM regardless of weight. 

The large and complex RUTF dosing used in treating SAM was questioned and several trials showed that reducing and simplifying the dosage does not adversely impact the efficacy of treatment [4,5,6,9]. It also appears that caregivers complement the child’s diet with family food regardless of the RUTF dose given [10]. Three clinical trials indicate that a gradual reduction from a high dose toward a lower dose of a nutritional product during the therapeutic path can be clinically non-inferior and cost-effective [4,6,9,11]. 

Recent research also explored the possibility of simplifying admission protocols for acute malnutrition by relying on the MUAC and edema only instead of requiring WHZ measurements [4,5,6]. The simplification of admissions criteria is based on research showing that raising the MUAC cut-off identifies children at the highest risk of mortality [12,13]. This can both save health workers time spent in measuring weight and height, and limit errors in z-score readings. Treatment by low-literate community health workers (CHWs) would also become easier with fewer steps and space for errors [14,15]. Enabling CHWs to deliver treatment was proposed as a solution to increase treatment coverage [16], which currently stands at <20% globally [17]. Treatment delivery by a CHWs reduces the distance from treatment sites [18], which was identified as one of the main barriers to accessing treatment [19,20]. Well-trained and supervised CHWs were shown to be capable of delivering good-quality malnutrition treatment in Mali [16] and treatment delivery by CHWs is now part of the Malian national protocol [21].

Another barrier that was frequently identified as preventing families from seeking treatment is unawareness of malnutrition [19]. To help caregivers identify malnutrition, it was proposed to train them in the use of and equip them with a MUAC tape [22]. Following evidence showing caregivers can be trained to correctly use the MUAC tape and search for edema [23,24,25,26], several countries adopted the Family MUAC approach as part of the national policy [22]. Considering that only the MUAC and edema are used to screen children in the community [27], having a single tool used for both screening and admitting children into treatment could simplify things, both for the caregivers and the healthcare providers. Caregiver’s screening children based on MUAC measurements would easily make the link between MUAC reading and treatment eligibility.

The ComPAS trial previously showed non-inferiority of a protocol where children were admitted to malnutrition treatment based on their MUAC measurement and treated with two daily sachets of RUTF for those with MUAC < 115 mm and one daily sachet of RUTF for those with 115 ≤ MUAC < 125 mm [4,28,29,30,31]. However, the study left several questions unanswered, including the effectiveness of this protocol in a routine setting and among some potentially more vulnerable sub-groups. Additionally, there is interest in looking at effectiveness when treatment is provided by CHWs and when families are trained in screening with the MUAC.

To build further evidence for the effectiveness of a simplified, combined malnutrition treatment protocol in a routine setting, we piloted it in rural Mali. The aim was to observe the characteristics of children admitted to treatment based on a MUAC < 125 mm and/or edema and their response to treatment. We also aimed to describe the admission characteristics by screener (caregiver versus CHWs versus formal health worker) and treatment results by treatment site (health facility versus community health site). In addition, we assessed the treatment response of several potentially more vulnerable subpopulations, such as those presenting with SAM in terms of both MUAC and WHZ, those treated as MAM but with WHZ < −3, and those with concurrent wasting and stunting. 

## 2. Materials and Methods

### 2.1. Study Design

This was an observational cohort study that described the response to simplified, combined treatment among children 6–59 months with MUAC < 125 mm and/or edema indicating acute malnutrition.

### 2.2. Study Setting and Population

The study was set in the Nara district in the southwest of Mali and was implemented throughout the 35 health areas of the district. The region is predominantly arid land [32] and the population is subsisting on small-scale farming and herding [33]. The region is less affected by insecurity and conflict compared with the northern and central regions. Food security in the region has been relatively stable since 2018 [34,35,36,37,38,39]. However, access to health care remains an issue, with 29% of the population in the district living over 15 km away from basic services [40]. In the region, only 48% of children 12–24 months of age are fully vaccinated [41] and the malaria prevalence is 22% among children under 5 years of age [41]. Acute malnutrition prevalence as measured by a WHZ < −2 was estimated at 7.6% in 2019 [42] and 6.4% in 2020 [43]. The International Rescue Committee (IRC) has been present in the area since 2015 supporting nutrition activities, including strengthening community screening activities and healthcare worker skills to manage acutely malnourished children.

### 2.3. Treatment and Measures Taken

The simplified treatment protocol was based on the protocol studied in the ComPAS trial [28] and included (1) admitting children based on their MUAC measure (<125 mm) or presence of bilateral edema, (2) treating children with MUAC < 115 mm and/or edema with 2 daily sachets of RUTF and children with a MUAC between 115 mm and 124 mm with 1 daily sachet of RUTF, (3) transitioning children admitted with MUAC < 115 mm or edema to receiving 1 sachet per day after 2 weeks with a MUAC ≥ 115 mm and an absence of edema, and (4) discharging children after 2 consecutive measures of MUAC ≥125 mm and absence of edema for 2 weeks. All children were followed up weekly at the treatment site. MUAC and weight measurements were taken at each visit, while height measurements were taken monthly and only at the health facility level. 

In addition to the nutritional treatment, children admitted with a MUAC < 115 mm or edema received a 7-day course of antibiotics (50–100 mg of Amoxicillin/kg/d) at admission, as per the international and national protocols for the management of SAM [21,27,44]. All children over 12 months of age and admitted to treatment received deworming (200–400 mg of Albendazole or 500 mg of Mebendazole) at the 1st follow-up visit as per the national protocols [21,45]. Malaria, diarrhea and acute respiratory infections were treated according to national integrated management of childhood illness guidelines [46] upon detection at any point during treatment. A discharge ration of 7 RUTF sachets was given to children upon recovery.

### 2.4. Implementation

All head nurses and nutrition focal points from the 35 functional health centers were trained (3 days of theory and 2 days of practice) in the simplified, combined protocol administration, in addition to the district management team. Admissions to the simplified treatment started in December 2018. Furthermore, a total of 38 CHWs were gradually trained to add malnutrition treatment according to the simplified protocol to their other integrated community case management activities. Training of the CHWs included 3 days of theory and 2 months of practical internship at a health facility for malnutrition treatment days before starting treatment at the community level. Five CHWs started providing treatment in 2019 and 33 more in 2020. Finally, women of reproductive age in the district were trained in measuring the MUAC of their children. The initial training sessions were done through CHVs in 2018 and a new wave of training sessions was then rolled out in 2020 by IRC trainers, together with CHVs. Throughout the pilots, UNICEF ensured the continued supply of the RUTF, and no stockouts were reported. 

### 2.5. Outcomes

The main outcome was the percentage of children that recovered. Secondary outcomes included defaulting, referral, non-response and death. Additionally, we calculated the mean length of stay (LOS), mean consumption of RUTF, mean MUAC gain velocity and mean weight gain velocity. 

### 2.6. Definitions

Recovery was defined as a MUAC ≥125 mm and no edema for 2 consecutive visits. Non-response was defined as not having attained recovery by 16 weeks of treatment. The recovery, defaulting, non-response and death percentages were calculated over these four exit categories as per the community-based management of acute malnutrition reporting guidance [47]. Inpatient transfers stayed in the program until attaining one of the 4 discharge criteria. Inpatient transfers included children referred for inpatient care due to medical complications, weight loss or negative appetite during treatment. Any patient out of the treatment system for more than 2 weeks was considered a defaulter.

The length of stay was calculated as the days from admission to discharge, with discharge being the last visit the child was cared for at the health facility. The weight gain velocity was calculated as the discharge weight—admission weight in grams divided by the admission weight in kilos and divided by the length of stay in days. The MUAC gain velocity was calculated as the discharge MUAC—admission MUAC in mm divided by the length of stay in weeks.

The initial screening of children was done either by a “health worker”, “CHW or community health volunteer (CHV)” or “caregiver”. Screening by a health worker meant that the caregiver did not report that their child had been screened at the community level but only once sought health services at the health center level. Screened by a CHW/V meant that the child had either been screened through campaigns or other active case finding in the community. Screened by a caregiver meant that the caregiver had been trained in malnutrition screening with a MUAC tape and searching for edema and had detected malnutrition and brought the child to care.

### 2.7. Data Collection

Individual treatment data were collected electronically for all children admitted to care with the help of the CommCare application and based on data noted on patient registries and individual patient cards at treatment sites. This data included the weekly anthropometrics of all admitted children and their respective discharge statuses. Morbidity data were not recorded. The currently analyzed data included all data from children admitted to treatment before the 1st of January 2022. Children that were still in treatment at

### 2.8. Data Analysis

Baseline characteristics and treatment outcomes of the study population are summarized as percent (n) for categorical variables or mean [SD] for normally distributed continuous variables and median [IQR] for non-normally distributed continuous variables. T-test were used for estimating statistical significance in recovery rates between sub-groups. All analyses were performed using STATA 15 (StataCorp, College Station, TX, USA). A complete checkup of data was performed prior to analysis to check for duplicates, outliers and missing data, and initial records were traced back if needed. A data quality review was also conducted by looking at the plausibility of data in terms of the MUAC distributions and changes during treatment. 

### 2.9. Ethics

The pilot study was approved by the national ethics committees of Mali (decision number 22/2018/CE-INRSP), as well as the IRC ethics committee (protocol number: H 1.00.025). All data used in the analyses were based on routine data collected by Ministry of Health staff upon treatment. No individual consent was used, as data collection was based on routine data collected by health workers. Communities were sensitized to the changes to the treatment protocol including changes in admission criteria and RUTF dosing, discharge criteria from treatment, delivery of treatment by CHWs and training sessions for caregivers to screen children using the MUAC and edema. 

## 3. Results

A total of 27 800 children were admitted to the simplified, combined treatment program (Table 1) between December 2018 and December 2021 (Appendix A). Around 35% of the children were admitted with a MUAC < 115 mm or edema, with only 0.1% with edema. The mean age of the children at admission was 15.1 months and the mean MUAC at admission was 108.2 mm for the children admitted with a MUAC < 115 mm or edema and 119.4 mm for children admitted with a MUAC between 115 and 124 mm. Among the children admitted with a MUAC < 115 mm or edema, 65.7% also had a WHZ < −3. Among the children admitted with a MUAC between 115 and 124 mm, 31.0% had a WHZ < −3 at admission and 70.0% had a WHZ < −2 at admission.

Most children were initially screened as malnourished in the community: 41.6% by CHWs or CHVs and 41.4% by the caregiver (Table 2). The majority (80.1%) of children were treated at the health center level, with 19.9% treated at the community health sites (Table 2). Admission anthropometry seemed somewhat higher among the children screened or treated at the community level. The mean MUAC was 114.5 mm upon admission among the children screened by a health worker compared with 115.5 mm among children screened by the caregiver and 115.9 mm among children screened by a CHWs or CHVs. The mean MUAC among children treated at the health center level was 115.2 mm compared with 116.5 mm among children treated at the community health sites. The mean WHZ was −3.0 among children screened by a health worker compared with −2.8 when screened by the CHWs, CHVs or caregiver. 

From December 2018 to December 2021, the overall recovery rate was 92.3% (Table 3), with 7.1% defaulting and <1% non-responses. Only 3.1% of children were referred to inpatient care during this period. We conducted a qualitative review of enumerator comments for patients that defaulted and did not respond. We found that the causes of defaulting included travel/relocation to another place, distance and sickness of the caregiver. Noted causes of non-responses were due to rejection of the RUTF by the child, presence of concurrent illnesses and RUTF ration sharing with other family members. Among children admitted with a MUAC < 115 mm or edema, 86.9% recovered. The mean length of stay was 39.6 days overall and 54.5 days among children with a MUAC < 115 mm or edema at admission. The overall weight gain velocity was 5.1 g/kg/d and 5.8 g/kg/d among those with a MUAC < 115 mm or edema at admission. Children with a MUAC < 115 mm and a MUAC between 115 mm and 124 mm at admission consumed on average 97 and 44 sachets of RUTF in the course of their treatment, respectively. Most children (84.4%) did not miss any visits during their treatment. Admission and treatment characteristics of children by treatment outcomes can be found in Appendix A.

The recovery rate exceeded 85% in all sub-groups studied (Table 4). Overall, 92.0% of children recovered when treated at the health facility compared with 93.5% cared for at the community level. Appendix A presents the p-values for t-test of difference in recovery rates between sub-groups. Appendix A presents the median length of stay in treatment by sub-groups, which ranged from 48 days to 63 days for sub-groups of children admitted with a MUAC < 115mm or edema and was 28 days in all sub-groups with a MUAC of 115–124 mm at admission. 

## 4. Discussion

This study showed that a simplified, combined treatment of children with acute malnutrition can be effective in a routine care setting as observed in this rural Sahelian context, where a total of 27,800 children were admitted to the treatment program over more than 3 years. All programmatic indicators exceeded the SPHERE standards [48] in all sub-groups previously identified as potentially more vulnerable or less responsive to treatment [5,9,28,29], including children with concurrent wasting and stunting, children with high weight at admission and older children. Treatment provided by CHWs resulted in similarly high recovery to when treatment was provided at the health facility level. This is the first paper to report the use of this simplified, combined protocol delivered outside health facilities, and the results suggest that the simplified protocol was easy to implement, even by non-formally trained care providers.

The high recovery and relatively short time to recovery that was observed in the current study are in line with observations from other programmatic studies conducted in Mali using the standard protocol [18,49]. In the ComPAS randomized controlled trial conducted in Kenya and South Sudan, the simplified, combined protocol was shown to result in a similar recovery rate and length of stay as the standard protocol [4]. The current findings strengthen the evidence that the program performance is as good when implementing the simplified, combined protocol, including for subgroups that could be expected to respond less well.

The quality of community health-worker-led treatment was previously tested in Mali and shown to result in high recovery rates [49]. We also observed high recovery rates among children treated by CHWs with the simplified, combined protocol. The fact that no height measurements were needed simplified the tasks for the CHWs. In the pilot area, the CHW delivery component of the program was only scaled up in mid-2020, which explained the low proportion of admissions (19.9%) to community health sites. When looking at the last 18 months of the program starting right after the scale-up of treatment by CHWs, up to 31% of children were admitted to treatment at the community health sites. Importantly, the recovery rate among children admitted to care at a community or health facility level was similarly high at 93.5% compared with 92.0% at the health facility level. 

Children admitted to care at the CHW sites seemed to have somewhat better anthropometric status compared with children admitted to care at the formal health center level. This is in line with what was reported previously [16,50] and what can be expected, and indicates that caregivers seek treatment sooner when a treatment site is closer to them. Enabling CHWs to easily adopt treatment activities by simplifying the protocol is one of the key advantages of simplified protocols to increase the coverage of treatment. 

Defaulting was low in this context. This is an important observation, as the simplified, combined protocol involves a smaller weekly RUTF ration compared with the standard protocol for some children with SAM [28,31]. The smaller RUTF ration could increase defaulting if the opportunity cost for seeking care supersedes the true and perceived value of the treatment received by the caregiver. This risk was not confirmed in the current context. Non-recovery was previously associated with distance from a health center [5,51]. In contexts with longer distances and potentially higher expenses for traveling and accessing care, defaulting could still be an issue. In the current pilot, the fact that CHWs were trained to provide treatment reduced the distance to treatment for 21% of the district catchment population. Still, 20% of the population remained at >15 km from treatment.

Certain sub-groups were previously identified as potentially responding less well to treatment [4,5,9,29,52] or generally more at risk of mortality [53,54]. Those include children with SAM according to both criteria [4,5], children with a severely low WHZ or MUAC [52], children with concurrent wasting and stunting [29,53,54], girls [5,55] and young children [5,9,52,56]. We showed that all of these groups responded well to the simplified protocol in a routine setting, as shown by the program indicators for these subgroups that exceeded the SPHERE standards. In this setting, the simplified, combined protocol resulted in good recovery of all children. 

The amount of RUTF consumed per child treated was 96 sachets for children admitted with a MUAC < 115 mm or edema. Unpublished programmatic data from the same context just a year earlier reported a mean consumption of 128 sachets of RUTF per child treated for SAM. In general, programs plan for approximately 150 sachets or 1 full carton of RUTF per child treated for SAM [57]. The ComPAS trial observed that the consumption of RUTF was decreased by 30–35% with the simplified protocol among children admitted with a MUAC < 115 mm compared with the standard protocol [4]. Even with the use of the higher-cost RUTF instead of the ready-to-use supplementary food to treat children with a MUAC between 115 and 124 mm, the overall program cost per child of the combined protocol was 12% lower than for standard care [4]. This study confirmed the potential for cost savings when using the simplified, combined protocol.

A previous study in Niger had observed that children screened as malnourished by caregivers came to treatment earlier than when detected by CHWs [23]. This was also observed in the current study, where the average MUAC at admission among children screened via Family MUAC (116.0 mm) seemed slightly higher than when screened by CHWs (115.4 mm) or at the health facility (114.6 mm). The mean MUAC at admission (115.5 mm) was in general lower compared with previous studies admitting children with a MUAC < 125 mm: 117 mm in the ComPAS trial in Kenya and South Sudan [4], 118.7 mm in the Optima study in Burkina Faso [5] and 121 mm in Sierra Leone [6].

Around 40% of children were reported to have been initially detected as malnourished by a caregiver, while 43% were detected by a CHW or CHV and 17% at a health facility. There is no reference or standard for the proportion of children that could be expected to be detected via the Family MUAC approach in contexts where this approach is being implemented. We could observe that the proportion fluctuated monthly throughout the three years of implementation from 20% to 70%, depending on recent screening and sensitization activities. 

In general, this pilot observed a higher recovery rate, higher weight gain velocity and lower length of stay than the ComPAS randomized controlled trial conducted in Kenya and South Sudan [4]. This may be explained by the differences in the study contexts. The ComPAS randomized controlled trial experienced low adherence to treatment in both study arms due to accessibility issues in both Kenya and South Sudan, a 6-month-long nurses’ strike in Kenya and the inability of caregivers in Kenya to take time off to bring their child to treatment visits [4]. In contrast, the current study was conducted in a rather stable rural context with high adherence to treatment, as observed through a very low proportion of children (<16%) ever skipping a visit. Treatment adherence was shown to be a key predictor of time to recovery, with one skipped visit associated with 3 weeks longer treatment time [58].

The main strength of this study was the high number of children (*n* = 27,800) included in the analysis. This enabled the investigation of several otherwise very small sub-groups and their response to treatment. All potentially vulnerable sub-groups previously identified responded well to the simplified treatment. Finally, we also include data from children admitted to treatment in the community health sites. This enabled us to observe the performance of the simplified treatment when provided by CHWs that gave similar outcomes as when delivered by formally trained healthcare workers.

The main limitation of the current study was the absence of a comparator group. The lack of a comparator group was by design, as the aim of this study was to document the use of the simplified, combined protocol in a routine setting. The initial ComPAS trial tested the effectiveness of the currently studied protocol in a randomized controlled trial and concluded that recovery was non-inferior in the simplified treatment group compared with the standard treatment [4]. Another limitation was that we relied on data collected by routine healthcare workers. However, we had robust supervision in place and thoroughly checked and cleaned the data before analysis, thus ensuring our confidence in the data collected.

## 5. Conclusions

This study aimed to observe the program performance of simplified, combined treatment when delivered by a routine health system. In conclusion, the results showed that the simplified, combined protocol resulted in a high recovery rate and low RUTF consumption per child. This simplified, combined treatment was also effectively delivered through CHWs. 

## Figures and Tables

**Table 1 nutrients-14-04923-t001:** Baseline characteristics of children admitted to treatment according to simplified, combined protocol.

Characteristic	N Not Missing	Admission MUAC and Edema Status		
		MUAC < 125 mm or Edema (All)	MUAC < 115 mm and/or Edema	115 ≤ MUAC < 125 mm and No Edema
Total, % (N)	27,800	100% (27,800)	34.9% (9710)	65.1% (18,090)
Boys, % (n)	27,800	46.2% (12,831)	44.9% (4358)	46.8% (8473)
Age in months, median [IQR]	27,800	12 [10,20]	12 [9,18]	13 [10,20]
Age group, % (*n*)	27,800			
<24 months		78.9% (21,936)	81.7% (7930)	77.4% (14,006)
≥24 months		21.1% (5864)	18.3% (1780)	22.6% (4084)
MUAC (mm), median [IQR]	27,800	117 [112,120]	110 [105,112]	120 [117,121]
Weight (kg), mean [SD]	27,800	6.7 [1.2]	6.0 [1.1]	7.1 [1.2]
Height/length (cm), mean [SD]	22,388	71.0 [6.6]	69.0 [6.6]	72.2 [6.4]
WHZ, mean [SD]	22,321	−2.9 [1.3]	−3.4 [1.3]	−2.5 [1.2]
WAZ, mean [SD]	27,772	−3.3 [1.2]	−3.9 [1.1]	−2.9 [1.1]
HAZ, mean [SD]	22,388	−2.3 [1.8]	−2.7 [1.8]	−2.0 [1.8]
WHZ category, % (*n*)	22,321			
<−3		43.6% (9731)	65.7% (5325)	31.0% (4406)
−3 ≥ to <−2		33.4% (7447)	23.5% (1904)	39.0% (5543)
≥−2		22.6% (5034)	10.4% (840)	29.5% (4194)
Presence of edema, % (*n*)	27,800	0.1% (28)	0.3% (28)	0 (0)

Abbreviations: HAZ, height-for-age z-score; MUAC, mid-upper-arm circumference; WAZ, weight-for-age z-score; WHZ, weight-for-height z-score.

**Table 2 nutrients-14-04923-t002:** Baseline characteristics of children admitted to simplified, combined treatment according to screening and treatment location.

Characteristic	Screened by			Cared at	
	Health Worker	CHW/V	Family MUAC	Health Center	Community Site
N (%)	4751 (17.1%)	11,551 (41.6%)	11,498 (41.4%)	22,267 (80.1%)	5533 (19.9%)
Boys, % (n)	49.3% (2340)	45.6% (5272)	45.4% (5219)	46.3% (10,318)	45.4% (2513)
Age in months, median [IQR]	12 [9,19]	12 [9,20]	12 [10,20]	12 [9,19]	13 [10,24]
MUAC (mm), median [IQR]	116 [110,120]	118 [113,120]	117 [112,120]	117 [112,120]	119 [114,120]
MUAC category, % (n)					
<115 mm	40.0% (1901)	31.9% (3685)	35.7% (4104)	36.6% (8139)	28.0% (1551)
≥115 mm	60.0% (2850)	68.1% (7866)	64.3% (7394)	63.4% (14,128)	72.0% (3982)
WHZ, mean [SD]	−3.0 [1.3]	−2.8 [1.3]	−2.8 [1.3]	−2.9 [1.3]	−2.6 [1.6]
WHZ category, % (n)					
<−3	48.1% (2051)	41.9% (3670)	43.1% (4010)	43.7% (9504)	38.3% (227)
−3 ≥ to <−2	32.2% (1373)	34.0% (2975)	33.3% (3099)	33.5% (7270)	29.8% (177)
≥−2	19.2% (820)	23.6% (2065)	23.1% (2149)	22.3% (4847)	31.5% (187)

Abbreviations: CHV, community health volunteer; CHW, community health worker; MUAC, mid-upper-arm circumference; WHZ, weight-for-height z-score.

**Table 3 nutrients-14-04923-t003:** Program outcome indicators for the simplified, combined treatment program in Mali.

Outcome	Admission MUAC and Edema Status		
	MUAC < 125 mm or Edema (All)	MUAC < 115 mm and/or Edema	115 ≤ MUAC < 125 mm and No Edema
Recovered, % (*n*)	92.3% (25,655)	86.9% (8435)	95.2% (17,220)
Defaulted, % (*n*)	7.1% (1985)	11.8% (1145)	4.6% (840)
Non-response, % (*n*)	0.4% (108)	0.9% (91)	0.1% (17)
Died, % (*n*)	0.2% (48)	0.4% (35)	0.1% (13)
Referred to inpatient care during treatment, % (*n*)	3.1% (864)	6.8% (664)	1.1% (200)
Length of stay (d), median [IQR]			
All discharges	35 [28,49]	50 [42,63]	28 [21,35]
Recovered only	35 [28,49]	56 [42,67]	28 [21,35]
MUAC gain velocity (mm/d), mean [SD]			
All discharges	0.3 [0.1]	0.4 [0.1]	0.3 [0.1]
Recovered only	0.3 [0.1]	0.4 [0.1]	0.3 [0.11]
Weight gain velocity (g/kg/d), mean [SD]			
All discharges	5.1 [2.9]	5.8 [2.8]	4.7 [3.0]
Recovered only	5.1 [2.8]	5.8 [2.7]	4.8 [2.7]
Number of RUTF sachets consumed, median [IQR]			
All discharges	49 [35,84]	91 [77,112]	42 [35,49]
Recovered only	49 [42,84]	98 [77,112]	42 [35,49]
Number of missed visits during treatment, % (*n*)			
None	84.4% (23,450)	81.6% (7919)	85.9% (15,531)
1 missed visit	13.5% (3765)	15.6% (1516)	12.4% (2249)
More than 1 missed visit	2.1% (585)	2.8% (275)	1.7% (310)

Abbreviations: MUAC, mid-upper-arm circumference; RUTF, ready-to-use therapeutic food.

**Table 4 nutrients-14-04923-t004:** Recovery from malnutrition in Mali following the simplified, combined treatment by sub-groups.

Subgroups	Recovery % (n)		
	MUAC < 125 mm or Edema (All)	MUAC < 115 mm or Edema	115 ≤ MUAC < 125 mm and No Edema
WHZ category			
WHZ < −3	89.7% (8729)	85.7% (4561)	94.6% (4168)
WHZ ≥ −3	93.9% (11,819)	88.5% (2459)	95.4% (9360)
Age group			
<24 months	93.0% (5451)	86.6% (1542)	95.7% (3909)
≥24 months	92.1% (20,204)	86.9% (6893)	95.0% (13,311)
Weight category			
≤7 kg	91.0% (16,193)	86.5% (7034)	94.8% (9159)
>7 kg	94.5% (9462)	88.6% (1401)	95.7% (8061)
MUAC category			
<110 mm	81.6% (2879)	81.6% (2879)	
≥110 mm	93.8% (22,776)	89.9% (5556)	95.2% (17,220)
WAZ category			
<−3	90.8% (14,636)	86.3% (6727)	95.0% (7909)
≥−3	94.4% (11,000)	89.4% (1689)	95.4% (9311)
Combined severe wasting and stunting			
WHZ <−3 and HAZ <−3	89.0% (7167)	85.2% (4108)	94.7% (3059)
WHZ ≥−3 and HAZ <−3	92.8% (4475)	88.0% (1466)	95.4% (3009)
WHZ <−3 and HAZ ≥−3	93.0% (1551)	90.0% (451)	94.3% (1100)
WHZ ≥−3 and HAZ ≥−3	94.5% (7260)	89.4% (966)	95.4% (6294)
Screened by a			
Health worker at a health facility	90.2% (4109)	84.6% (1545)	94.0% (2564)
CHW or CHV	93.2% (9822)	88.8% (3380)	95.7% (6442)
Caregiver (family MUAC)	92.3% (10,624)	86.0% (3167)	95.3% (7457)
Cared for at a			
Health facility	92.0% (20,484)	86.7% (7068)	95.1% (13,416)
CHW site	93.5% (5171)	88.0% (1367)	95.6% (3804)

Abbreviations: CHV, community health volunteer; CHW, community health worker; HAZ, height-for-age z-score; MUAC, mid-upper-arm circumference; WAZ, weight-for-age z-score; WHZ, weight-for-height z-score.

## Data Availability

Data may be accessed and downloaded at https://doi.org/10.5281/zenodo.7086233 (accessed on 16 September 2022).

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
