# Peer review of "Effectiveness of Acute Malnutrition Treatment at Health Center and Community Levels with a Simplified, Combined Protocol in Mali: An Observational Cohort Study"

_nutrients, 2022, doi:10.3390/nu14224923_

Round 1

Reviewer 1 Report

General
·      The paper reports important data which are very worth to be published.
·      Still, the authors may consider to present their data not in the highly aggregated format only and with more details, also on the ‘defaulters’ and ‘non-responders’, the length of observation after completing the rehabilitation, and relapses.
·      The term ‘observational cohort study’ does not exclude the utilization of statistical methods. Some differences between the subgroups should be checked for statistical significance.
·      A limitation of the study is the age-range of the children studied. This is not a critique, but the conclusions may not hold true for children at lower and higher ages. Especially, the utilization of the fixed MUAC-cut-offs is critical in older children.

Specific:
Results
- Although its not a randomized trial a modified Consort-Flow Diagram would be informative for comparing the subgroups screened/ treated differently and the children getting out of the main data stream. The mix of the different management aspects (screening and care) makes it a bit difficult for the reader to understand the effects.

Table 1
-
The headline inscription is very unclear: ‘<125 mm or edema’, ‘<115 mm or edema’ and ‘115 to  <125 mm or edema’ – does the first column include data from one of the others ?

- N-data in the lines ‘Age Group’ of the subgroups don’t add up to 27601 ! The authors may check the N-data in all tables as there are more inconsistencies.
Table 2
- In the line ‘N’ more children get ‘cared’ than ‘screened’
à check data and consider dividing the table in 2.1 screening and 2.2. caring for more clarity.
Line 231
- The phrase ‘Only 3.1% of children were referred to inpatient care at some point in treatment.’ Should be specified: How many were sent to IC initially, how many at what times ? What were the causes of ‘default’ and ‘non-response’ ?

Line 232
-
If 93.7% recovered only in the subgroup <115mm : what happened to the 6.3% ?

Table 3
- The authors may indicate how the groups ‘defaulted’,‘non-response’, ‘died’ and ‘referred’ relate to each other. Are the same children in more than one subgroup listed ?

Table 4            see comment on Table 1
Table 1 Suppl.
Needs to be checked for the definition of the subgroups too.

Author Response

General

- The paper reports important data which are very worth to be published.

Author We thank the reviewer for providing positive and valuable comments on the manuscript. We provided the responses to the specific comments below.

- Still, the authors may consider to present their data not in the highly aggregated format only and with more details, also on the ‘defaulters’ and ‘non-responders’, the length of observation after completing the rehabilitation, and relapses.

Author:  We appreciate this suggestion. We’ve added a supplemental table describing the population of  non-responders and defaulters as well as their discharge statistics. This analysis adds detail on a small portion of the population, and we feel that including these in the main body of the paper would distract from the primary message.

-  The term ‘observational cohort study’ does not exclude the utilization of statistical methods. Some differences between the subgroups should be checked for statistical significance.

Author:  Thank you for this suggestion, we have added statistical tests of difference in recovery rates across subgroups in the supplemental tables. Our primary aim in this paper was to estimate effectiveness across a range of populations of interest, not necessarily to determine differences in effectiveness across these populations. Those differences are ultimately both intuitive and well understood (For example: we would expect very precise estimates of recovery to be lower among children with MUAC<110).  We’re happy to present these though we think the more interesting and central point is that, despite significant but small magnitude differences, all groups recovered very well. If there are additional tests that the reviewer had in mind, we are happy to consider those. 

- A limitation of the study is the age-range of the children studied. This is not a critique, but the conclusions may not hold true for children at lower and higher ages. Especially, the utilization of the fixed MUAC-cut-offs is critical in older children.

Author:  Thanks to the reviewer for this comment. The majority of children in this study are between 6 to <24 months. This reflects the reality of most treatment programs globally, as wasting (by any measure) is most common in this age range. Because of the large sample size, a considerable number of children between 24 to 59 months are also included in our analyses (n=5,640) and these children showed high recovery (97%).

Specific

Results

- Although its not a randomized trial a modified Consort-Flow Diagram would be informative for comparing the subgroups screened/ treated differently and the children getting out of the main data stream. The mix of the different management aspects (screening and care) makes it a bit difficult for the reader to understand the effects.
Author:  We would like to thank the reviewer for this comment. We agree to add a modified Consort-flow diagram (figure 1) in the supplement clarifying any points of attrition between the patients identified as eligible and the final sample size (27,601 patients).

Table 1

- The headline inscription is very unclear: ‘<125 mm or edema’, ‘<115 mm or edema’ and ‘115 to  <125 mm or edema’ – does the first column include data from one of the others ?

Author:  We thank the reviewer for this question. The first column ‘<125 mm or edema’ represents all the children admitted to treatment according to the simplified, combined protocol.  This column combines data from both second and third columns. We have also clarified that the last group ‘115 to <125 mm does not include any edema case.  In order to further clarify the categories, we have added (All), (SAM), (MAM) to the headers.

- N-data in the lines ‘Age Group’ of the subgroups don’t add up to 27601 ! The authors may check the N-data in all tables as there are more inconsistencies.

Author:  We made the appropriate checks in the entire manuscript. On the specific lines ‘Age Group’, the subgroups in the first column ‘<125 mm or edema’ add up to 27601. For the remaining columns, we confirm that the line of column ‘MUAC<115 mm or edema’ and ‘MUAC<115 to <125 mm also add up to 27601.  This may be due to the confusion around the column headers addressed above.

Table 2

- In the line ‘N’ more children get ‘cared’ than ‘screened’ à check data and consider dividing the table in 2.1 screening and 2.2. caring for more clarity.

Author:  We split the table 2 into two tables, 2.1 and 2.2 as recommended. After checking the data, we made appropriate corrections on both tables. 

Line 231

- The phrase ‘Only 3.1% of children were referred to inpatient care at some point in treatment.’ Should be specified: How many were sent to IC initially, how many at what times ? What were the causes of ‘default’ and ‘non-response’ ?

Author:  We added some missing details into the phrase on page 6 line 239 to clarify that the 3% of children referred to inpatient care concerned the overall observed period i.e., from December 2018 to December 2021. We also describe that possible causes of defaulting may be due to travel /relocation to another place, distance, and sickness of the caregiver. The possible causes of non-response included rejection/non consumption of the RUTF by the child, presence of concurrent illnesses, RUTF ration sharing with other family members.

Line 232

- If 93.7% recovered only in the subgroup <115mm : what happened to the 6.3% ?

Author:  We added some missing details into the phrase on page 6 line 239-244 to clarify that the 3% of children referred to inpatient care concerned the overall observed period i.e., from December 2018 to December 2021. The default was due to travel /relocation to another place, distance, sickness of the caregiver. The cause of non-response included rejection of the RUTF by the child, presence of concurrent illnesses, RUTF ration sharing.

Table 3

- The authors may indicate how the groups ‘defaulted’,‘non-response’, ‘died’ and ‘referred’ relate to each other. Are the same children in more than one subgroup listed ?

Author:   While a child may fall into multiple categories, we have prioritized the categories to make the outcomes unique and mutually exclusive. The order of priority is died, defaulted, non-response, cured. For example, if a child died during treatment, regardless of other circumstances, that patient is marked as died. If a child was cured, but also missed two consecutive visits during treatment, they would be uniquely marked as defaulted. Referred to inpatient care during treatment, is not considered an outcome, therefore patients that were treated as inpatients could also be one of the other four categories.

Table 4

- see comment on Table 1

Author:  Upon review we still believe that these categories add up correctly and we have similarly addressed the confusion around headers that was discussed above.

Table 1 Suppl.

- Needs to be checked for the definition of the subgroups too.

Author:  Upon review we did find that 15 cases were erroneously excluded from this analysis, this has been resolved.

Reviewer 2 Report

The paper is an fair overview of the current situation, but without a comparison group the results do not say much. It's about proving something that (we already know) helps and works. The methodology is basic and simple. On the other hand, the sample is really large and representative.

I recommend publication, having few questions/ suggestions regarding methods and results for the authors to improve clarity:

- please check the distribution. Some variables do not appear to be normally distributed. If this is true, the more appropriate notation is the median (interquantile rank).

- the record mean +- SD is inappropriate or misleading. More suitable is the mean [SD].

Author Response

The paper is an fair overview of the current situation, but without a comparison group the results do not say much. It's about proving something that (we already know) helps and works. The methodology is basic and simple. On the other hand, the sample is really large and representative.

Author:  We thank the reviewer for providing valuable comments on the manuscript. We provided the responses to the specific comments below.

I recommend publication, having few questions/ suggestions regarding methods and results for the authors to improve clarity:

  • please check the distribution. Some variables do not appear to be normally distributed. If this is true, the more appropriate notation is the median (interquantile rank).

Author:  This is a helpful note - we have conducted the Shapiro-Wilk and Shapiro-Francia tests for normality, while some variables do reject normality, the test also estimates that the samples are large enough for the Central Limit Theorem to apply. We also note that extreme values are cleaned as per WHO nutrition guidelines, and thus outliers should have a minimal effect on our distributions.

the record mean +- SD is inappropriate or misleading. More suitable is the mean [SD].

Author:  We made the appropriated corrections in the entire manuscript.

Round 2

Reviewer 1 Report

In line 204 the number 27,000 should be corrected into 27,601 if I understand it right.

Otherwise, the answers to my comments are satisfying the concerns.

Author Response

In line 204 the number 27,000 should be corrected into 27,601 if I understand it right.

Author:  Thank you very much for your comment, the number 27,000 has been corrected into 27,601.